# The Healthy Food Environment Policy Index in Poland: Implementation Gaps and Actions for Improvement

**DOI:** 10.3390/foods11111648

**Published:** 2022-06-02

**Authors:** Piotr Romaniuk, Krzysztof Kaczmarek, Katarzyna Brukało, Elżbieta Grochowska-Niedworok, Karolina Łobczowska, Anna Banik, Aleksandra Luszczynska, Maartje Poelman, Janas M. Harrington, Stefanie Vandevijvere

**Affiliations:** 1Department of Health Policy, Faculty of Health Sciences in Bytom, Medical University of Silesia in Katowice, Piekarska Street 18, 41-902 Bytom, Poland; promaniuk@sum.edu.pl (P.R.); kbrukalo@sum.edu.pl (K.B.); 2Department of Human Nutrition, Faculty of Health Sciences in Bytom, Medical University of Silesia in Katowice, Jordana Street 19, 41-808 Zabrze, Poland; grochowska-niedworok@pwsz.nysa.pl; 3Department of Psychology in Wroclaw, SWPS University of Social Sciences and Humanities, Ostrowskiego Street 30b, 53-238 Wroclaw, Poland; karolina.lobczowska@gmail.com (K.Ł.); abanik@swps.edu.pl (A.B.); aluszczy@uccs.edu (A.L.); 4Lyda Hill Institute for Human Resilience, University of Colorado at Colorado Springs, 1420 Austin Bluffs Pkwy, Colorado Springs, CO 80918, USA; 5Consumption and Healthy Lifestyles, Wageningen University & Research, P.O. Box 8130, 6700 EW Wageningen, The Netherlands; maartje.poelman@wur.nl; 6School of Public Health, University College Cork, T12 XF62 Cork City, Ireland; j.harrington@ucc.ie; 7Department of Epidemiology and Biostatistics, School of Population Health, University of Auckland, Auckland 1142, New Zealand; stefanie.vendevijvere@sciensano.be

**Keywords:** healthy food environment, food policy, policy evaluation

## Abstract

Background: Poland is facing the growing problem of overweight and obesity in the population, which makes it necessary to conduct a thorough assessment of the existing food environment policies. The aims of the study were: (1) to depict the strength of healthy food environment policies in Poland and identify implementation policies and infrastructure support gaps; (2) to identify and prioritise improvement policies, taking into account their importance, achievability and equity. Methods: We used the Healthy Food Environment Policy Index (Food-EPI). An experts’ panel rated Polish policies and infrastructure compared to international best practices and developed a list of recommended improvement actions addressing both components. Results: eight of the twenty-two policy and four of the twenty-two infrastructure indicators achieved the “no/very weak policy” result. Another four policy and five infrastructure indicators were considered “weak”. Another seven and eight indicators, respectively, were assessed as “moderate”. Among the identified actions, the highest priority was given to a food labelling system and training for persons involved in nutrition in schools. Conclusions: The Polish healthy food environment has been assessed as very weak or weak in most aspects. The infrastructure was assessed as slightly better compared to the policies domain, with more indicators receiving the “moderate” score.

## 1. Introduction

Like most developed countries, Poland faces a rising burden of obesity and diet-related non-communicable diseases (NCDs) [1,2]. In 2014, the prevalence of overweight for men and women was 62% and 46%, respectively, and the prevalence of obesity was 18% and 16%, respectively [2]. As the obesity epidemic is rising, it has been estimated that in 2025, 26% of women and 30% of men in Poland will be living with obesity [3]. Excess body weight remains an important risk factor for the development of numerous metabolic diseases, including type 2 diabetes, lipid abnormalities, arterial hypertension and other cardiovascular diseases, as well as some oncological diseases, with all of these health issues being widespread in the Polish population and among the main causes of death [3,4,5,6,7,8,9]. In the context of the epidemiological trends in obesity prevalence, developing a coherent, evidence-based and consistent overarching vision of food policy appears to be a crucial task for public governance [10].

Over the past two decades, food policy in Poland has been considered inconsistent and unconsolidated [10]. There are only a very limited number of studies analysing relevant national policies, but they point to a significant number of gaps, confirming previously expressed views on the overall state of food policy. While there are some solutions in place in regard to food-based guidelines addressing the general population’s dietary behaviours, nutrient intake and consumer choices [11,12], school nutrition [13] and food security [14], Polish policy does not provide a sufficient response addressing the dietary-related diseases in areas such as the monitoring and evaluation of behavioural change, food reformulation, consumer awareness and the setting of national nutrition targets [12,15].

A number of challenges and difficulties are associated with implementing and evaluating food policies [16]. These methodological limitations and the lack of reliable and up-to-date data are barriers to the development of food policy, and they also affect the political and public willingness to make it an important subject of the political agenda. In food policy and its scientific assessment, one challenge is the complexity of the food environment. Attempts have been made to analyse it more narrowly, for example, in a specific setting (e.g., food policy in schools) or specific population groups (e.g., the elderly population). However, doing so makes it impossible to present a complex picture of the food policy in the country, assess or investigate the interdependencies between individual dimensions of the food policy environment or provide clear and comprehensive recommendations for future actions to address key issues.

Our study is an effort to address these deficiencies of food policy in Poland by applying the Healthy Food Environmental Policy Index (Food-EPI) [17] to evaluate the implementation of food policy to improve food environments and the extent of its implementation.

The two basic aims of this study are as follows:To depict the overall strength of healthy food environment policies in Poland and to identify basic implementation gaps in terms of policies and infrastructure support.To identify and prioritise healthy food environment policies in Poland, taking into account their importance, achievability and equity.

## 2. Materials and Methods

The study was a part of Work Package 1 (WP1) of the Policy Evaluation Network (PEN) project (https://www.jpi-pen.eu/, accessed on 3 May 2022), funded by the Joint Programming Initiative Healthy Diet for Healthy Living (JPI-HDHL). To achieve the assumed study aims, we applied the Food-EPI [17], originally developed by the International Network for Food and Obesity/Non-Communicable Diseases Research, Monitoring and Action Support (INFORMAS) [18]. A review of existing tools indicated that Food-EPI is the most comprehensive instrument, allowing for evaluation of the food environment in any given country. Applying the same tool in countries involved in the project enables international comparisons and identification of common gaps existing in food policies and infrastructures.

The study was evaluated by the Biomedical Commission at the Medical University of Silesia in Katowice, which issued a waiver allowing to conduct the study. As stated in the Commission’s position No. PCN/0022/KB1/79/21, the study was not considered a medical experiment involving human subjects.

### 2.1. Study Procedure

The process of developing the Polish version of the Food-EPI consisted of five stages (see Figure 1).

#### 2.1.1. Step 1: Reviewing the Tool and Its Indicators

The Food-EPI was developed to assess the extent of implementation of national/public sector food and nutrition-related policies and to identify and prioritise policy actions for creating healthy food environments. The tool consists of “best practice” indicators across thirteen domains, of which seven are food policy-related (food composition, food labelling, food promotion, food prices, food provision, food retail, and food trade) and six are infrastructure-related (leadership, governance monitoring and intelligence, funding and resources, platforms for interaction, and health in all policies) (see Figure 2). The domains are subsequently divided into individual good practice indicators for evaluation [18,19]. Prior to the application of the Food-EPI to food policies in participating PEN countries (Ireland, Norway, Poland, The Netherlands and Germany), all Food-EPI indicators were reviewed by participating PEN partners. The changes applied to the original Food-EPI tool were assumed to adjust it to the context specific to European countries. After the consultation procedure was completed, it was agreed to apply the final list of 44 indicators (compared to 47 indicators in the original tool) and 12 domains to conduct an assessment in five different European countries. (See Appendix A).

#### 2.1.2. Step 2: The Catalogue of International Best Practices

The second step of the study was to develop a catalogue of international best practices by all PEN partners involved in the Food-EPI study. The basis for the catalogue was the benchmarks previously developed by INFORMAS in August 2017. The final version of the catalogue was approved by all study partners in November 2019. For details, see Appendix A.

#### 2.1.3. Step 3: National Evidence Document

For each of the Food-EPI indicators, evidence for the implementation of policies in Poland was collected by three researchers (PR, KK, and KB) who reviewed official government documents and legal acts [11,14,20,21,22,23,24,25,26,27]. All identified policies with a potential influence on food environments were assigned to one of the respective Food-EPI domains. The document was compiled between October and December 2019 and then sent for approval and validation by government officials representing the Ministry and the Chief Sanitary Inspectorate. The final version was prepared to constitute a basis for the further steps of the study. All the data included in the evidence document were collected from Polish sources and in the Polish language. Due to the fact that the document was intended to be used solely by Polish experts participating in the study, it was not translated into English. (See Appendix A).

#### 2.1.4. Step 4: Online Rating Survey

Between March and August 2020, an online rating survey took part. The study participants were an expert panel consisting of independent experts representing diversified sectors, especially the academic and research sector, non-governmental organisations (NGOs) providing food-related and health-related activities, and practitioners in the field of nutrition and chronic disease prevention or treatment. The list of experts included specialists identified by the research team based on their scientific achievements (reflected in their publication output) or specialists recognised in Poland for their achievements in the field of food policy development and implementation. The list of experts has been reviewed and approved by the leaders of the PEN Project Work Package. Out of 63 invited experts, 21 responded by completing the survey questionnaire. The participants were informed of the aims of the survey and the voluntary nature of their participation in the study. They were given the opportunity to complete the questionnaire anonymously and to give contact details to obtain formal confirmation of their involvement in the study. The questionnaire was prepared using Google Forms and consisted of questions referring to each of the Food-EPI indicators. Each question was accompanied by an information package presenting the international best practices, in accordance with the catalogue developed in step 2 of the study, as well as information on the identified evidence of the implementation of a given policy in Poland. The experts assessed national policies against the international best practices for each indicator using a percentage-based scale translated into a five-point Likert scale, where 1 = very weak policy (0–20% implementation), 2 = weak policy (30–40% implementation), 3 = moderate policy (50–60% implementation), 4 = strong policy (70–80% implementation), and 5 = very strong policy (90–100% implementation). A “cannot rate” option was also given for selection, as well as the opportunity to comment on any of the indicators or policies. For each policy-related indicator, the participants were asked to assess whether its implementation would impact socio-economic inequalities. The rating was performed on a 5-point Likert scale, where 1 = a significant increase in inequalities, 2 = a moderate increase in inequalities, 3 = no impact, 4 = a moderate reduction in inequalities, and 5 = a significant reduction in inequalities.

#### 2.1.5. Step 5: Identification and Prioritisation of Actions to Improve the Food Environment

On 15 September 2020, a workshop was organised for the study participants with the aim of identifying actions to improve the Polish food environment. Invitations were sent to the experts who took part in the previous steps of the study, resulting in eleven positive responses. In addition to the invited experts, three members of the research team facilitated the workshop (PR, KK, and KB). Due to restrictions caused by the COVID-19 pandemic, the workshop was organised online using a videoconferencing platform. The workshop consisted of two main stages. The first was discussion in two subgroups that worked on the development of proposals for actions with regard to policies and infrastructure in line with the domains and indicators of the Food-EPI. It was followed by a plenary discussion and agreement on the final version of the catalogues of action for both areas. After the real-time workshop, the research team synthetised the plenary discussion results in the form of two catalogues of proposed actions, representing recommendations developed during the meeting. These catalogues were sent for approval to the workshop participants. After adjusting the catalogues based on the revision requests received, the final list approved by the workshop participants consisted of 15 actions with regard to policies and 15 actions with regard to infrastructure. These two catalogues were then subject to a prioritisation procedure, which was held asynchronously. The workshop participants received both catalogues along with the prioritisation questionnaire and were asked to rank the policy actions separately for three dimensions: importance, achievability and equity. In the case of infrastructure actions, the ranking was performed for two dimensions: importance and achievability. “Importance” was defined as the need, impact of the action, and other positive or negative effects of the action. “Achievability” referred to the feasibility, acceptability, affordability and efficiency of the action. “Equity” was defined as the impact of the action on socio-economic inequalities in dietary intake and health and on individual consumer choices. Additionally, the study participants were given an opportunity to weight the prioritisation dimensions individually for each of the actions in the catalogue, using 100 percentage points to be distributed between relevant dimensions.

### 2.2. Data Analysis

For the healthy food environment and infrastructure support indicators, the average level of implementation was calculated based on the mean values scored by 21 experts who completed the Food-EPI questionnaire. The ranking of actions in the prioritisation step was established by calculating the mean value of scores attributed to individual actions by members of the expert panel (with each expert assigning from 1 to 15 points). When calculating the ranking with all dimensions combined, we summed the mean values of the scores for each dimension, which were additionally modified with the weight attributed to a given dimension by each individual rater.

We applied the bootstrapped Krippendorff’s alpha test to verify inter-rater reliability. The coefficient value was calculated using R software v. 4.0.4. All the remaining calculations were performed using MS Excel Professional Plus 2019.

## 3. Results

### 3.1. Evaluation of the Extent of Implementation of Healthy Food Environment Policies in Poland

#### 3.1.1. Overall Assessment of the Policy- and Infrastructure-Related Domains

An invitation to take part in the Food-EPI survey was sent to 63 experts in Poland. The response rate was 30%, with a total of 21 questionnaires that were finally returned. Among the respondents, 16 participants were professionally linked with academic institutions, two represented health care providers, two represented local public administration units, and four were linked to third sector organisations. Since double entries were allowed, the sum of these numbers is not 21.

The results for the assessment of the Polish health food environment in the policy-related domains are presented in Figure 3.

Out of the 22 indicators evaluated, we obtained a “no/very weak policy” result for eight (36%) indicators. In the case of four indicators (18%), the result was “weak policy”, while for seven (32%) indicators, we obtained a “moderate policy” result. The remaining three indicators (14%) were evaluated as “strong policy”, and no indicator scored the highest rating. The best ratings were observed for indicators related to restrictions on the promotion of unhealthy food in places of collective stay of children, such as schools or kindergartens. The “strong policy” rating also appeared in the case of food subsidies for health foods and policies in schools to promote healthy food choices. These policies appear in practice, e.g., in the form of free fresh fruit or vegetables or milk products distributed for free among children [28,29]. The food retail domain was entirely assessed as showing no or very weak policies in Poland.

The Krippendorff’s alpha coefficient value for policy-related domains was 0.546 (95% confidence interval (CI) = 0.522–0.568), which shows a moderate level of agreement between raters.

The results of the assessment of the Polish health food environment in the infrastructure-related domains are presented in Figure 4.

The overall result of the evaluation of the infrastructure-related indicators is slightly better than that of policies. Of the 22 indicators in this group, only four (18%) received the lowest “no/very weak policy” rating, five (23%) were assessed as “weak policies”, eight (36%) were rated as “moderate policy”, and five (23%) received “strong policy” ratings. As was the case in the policy-related domains, no case of “very strong policy” was registered. The Krippendorff’s alpha coefficient value for the infrastructure-related domains was 0.403 (95% CI = 0.379–0.428), which means that the level of agreement was close to moderate.

#### 3.1.2. Assessment of the Impact on Socio-Economic Inequalities

The study participants then assessed a set of policy-related indicators with regard to how the policies in a given area may impact socio-economic inequalities. For six indicators (27%), namely, restricting unhealthy food promotion to children (social media), restricting unhealthy food promotion where children gather, food subsidies to favour healthy foods, policies in schools to promote healthy food choices, food-related income support for healthy foods, and support and training systems (public sector), the result was a “small reduction”. For the remaining 16 indicators, no impact on health inequalities appeared to be expected by the experts. However, since the Krippendorff’s alpha coefficient value for this group of questions was 0.019 (95% CI = −0.019–0.056), we observed no agreement between raters, and drawing any clear conclusions with regard to this issue is not possible, especially because many respondents marked the “cannot rate” answer as their choice.

### 3.2. Identification and Prioritisation of Actions to Improve Poland’s Healthy Food Environment

In the subsequent stages of the study, we identified a catalogue of actions to be recommended to the Polish government that would improve the national food environment in terms of policy and infrastructure support. For each of the two Food-EPI components (the policy and infrastructure domains), 15 actions were identified. Each expert ranked the identified action on a scale ranging from 15 (highest priority) to 1 (lowest priority) separately for each dimension. Table 1 presents the prioritised policy-related actions. The final order combines the prioritisation procedure for all three dimensions (importance, achievability, and equity), with the weights attributed to individual dimensions by the raters included in the calculation. The higher the position on the list is, the higher the priority of the action. The action labels are taken from the original list included in the questionnaire sent to the expert panel.

#### 3.2.1. Policy Actions

The highest priority has been attributed to the food labelling system, which mainly refers to the implementation of simple labelling systems, such as Nutriscore, followed by informational policy related to healthy food regulations and the change in legal regulations related to school meals. The lowest priority was given to restrictions on the density of fast-food bar restaurants in the public space. In total, one-third of the actions were related to the food provision domain, and one-fifth were related to the food retail domain. 

Figure 5 presents the actions with detailed results for importance and achievability combined. For the scores, no weighting was applied.

In this context, the implementation of a simple food labelling system was perceived not only as a priority action but also as the action that is the most important and the most achievable. In general, actions were assessed coherently in all three dimensions, including equity. Different situations were observed for several actions. For example, taxation policy (action No. 5) was considered very important but difficult to achieve. On the other hand, the rules for eating school meals (action No. 2) and the labelling of menus in restaurants (action No. 8) were assessed as highly achievable but as being of low importance. Importantly, the inter-rater agreement with regard to prioritisation was fair, with the Krippendorff’s alpha coefficient value being the highest in the case of scores for importance (α = 0.385, 95% CI = 0.327–0.441), followed by achievability (α = 0.246, 95% CI = 0.175–0.315). As was the case for socio-economic inequalities in the Food-EPI questionnaire, the level of agreement with regard to the equity dimension was lower (α = 0.156, 95% CI = 0.082–0.228).

#### 3.2.2. Infrastructure Support Actions

Table 2 presents the infrastructure-related actions in order resulting from the prioritisation procedure. The final order combines the prioritisation procedure for both dimensions (importance and achievability) with the weights attributed to individual dimensions by the raters included in the calculation. The higher the position on the list is, the higher the priority of the action assigned by the experts. The action labels were taken from the original list included in the questionnaire sent to the expert panel.

The highest priority was attributed to the implementation of healthy eating training sessions for people responsible for feeding children, followed by creating a system of promoting healthy eating habits in the media and including dietitian services in the basket of services covered by public health insurance. The lowest priority was attributed to regulations regarding food certification systems. Overall, domain No. 2 was addressed most frequently in the actions proposed by the expert panel, and it also appears most frequently in those actions that received the highest priority. 

Figure 6 presents the actions with detailed results for importance and achievability combined. For the scores, no weighting was applied.

We observed much higher diversity in terms of the scores for importance and achievability than in the case of policy support actions. Action No. 1 was assessed as having visibly higher importance than any of the other actions, but it had one of the lowest results in terms of achievability. In turn, action No. 6 obtained a relatively high score in terms of achievability but the lowest score in regard to importance. The overall level of agreement among the experts when prioritising the actions was somewhat lower than in the case of policy-related actions for importance (α = 0.217, 95% CI = 0.144–0.288), but it was slightly higher for achievability (α = 0.276, 95% CI = 0.208–0.342).

## 4. Discussion

Our study showed a number of gaps in food policies and infrastructure in Poland. In the case of policies, the largest deficiencies were found in the domains related to food promotion, food provision, and food retail. In the food promotion domain, the lowest score was given to “restricting unhealthy food promotion to children on social media” and “restricting unhealthy food promotion to children on packaging”. This observation seems to be in line with studies on public health-related informational policy in Poland, where it was found that public authorities neither utilise social media for health promotion [30] nor intend to regulate it [31], despite the evidence that its impact may be of a considerable scale [31]. In Poland, social media themselves are perceived as a threat to safety and as a potential source of peer violence [32] and not as a proper communication channel for health promotion, which may result in the consolidation of harmful eating habits.

The restrictions on packaging appear to be a similar case. The information located on product packaging has a considerable impact on consumer decisions [33]. This is especially important in the case of children, who are more susceptible to manipulation by the message associated with a product [34]. Although the issue of labelling is partly regulated at the EU level [35], the identified policy gap appears to be especially significant due to the importance of this regulation in terms of eliminating the negative impact on consumer behaviour. Additionally, it is more acceptable than any other consumption-related intervention, which enhances its potential in terms of political feasibility.

Interestingly, similar results have been obtained in Germany and The Netherlands, where application of the Food-EPI study revealed even more gaps existing in terms of food promotion, but with an equally low score in the case of restrictions related to social media and packaging [36,37]. Somewhat better results have been observed in New Zealand and Canada, although the level of implementation of policies addressing these domains was still assessed as low there [38,39].

In the food retail domain, all indicators received the lowest scores, which confirms that Polish public food policies do not propose any solutions aimed at retailers’ promotion of healthy consumer choices. While the large retail chains implement their own policies [40], no such policies exist in the case of small local groceries, despite the rising expectations on the consumer side regarding the availability of healthy food alternatives. In addition to health benefits, this change in consumer preferences potentially brings positive economic outcomes by developing a model of sustainable consumption [41]. While a friendlier institutional environment for direct sales of locally produced food was implemented in 2016–2017 [42], there is a lack of solutions that would improve its availability to a wider population, as direct sales are mainly based on social, highly trust-dependent networks. Due to historical and cultural conditioning, these networks remain weak in Poland, which limits the availability of products on the market and negatively affects equity in access. Nevertheless, nearly half of all Poles (49%) fulfil their consumption needs in discount stores [43]. Local marketplaces are among the least popular places for grocery shopping (preferred by as many as 3% of consumers [43]), and the price of products is still the dominant factor in consumer decisions.

Furthermore, in this case, the result shows similarities with a number of other Food-EPI-based studies in other countries, including New Zealand and Germany, although in the first of the countries mentioned, a somewhat better result was observed with regard to the indicator assessing promotion of relative availability of healthy foods in food service outlets. A slightly better but still relatively low score for this domain has been observed in Japan and Canada [36,39,44].

In the case of infrastructure-related domains, we found that 4 out of 22 indicators were assessed with the lowest score. These domains were priorities for reducing health inequalities (domain: leadership), the use of evidence in food policies and transparency in the development of food policies (domain: governance) and assessing the public health impacts of non-food policies (domain: health in all policies). The indicator addressing health inequalities was somewhat better scored in The Netherlands, Canada and Germany, while New Zealand and Japan have been assessed with considerably better scores [36,37,38,39,44]. The result referred to the use of evidence in food and nutrition policies is especially worrisome in Poland, as it was assessed considerably better in other countries, especially The Netherlands and Japan, with the latter one obtaining the highest possible score. Some gaps, although less critical, with regard to transparency in the development of food policies were identified in Germany, while again Japan, along with New Zealand may be considered benchmarks here, with the highest possible score obtained in respective studies [36,38,44]. Finally, in the case of the last indicator with the lowest score, a similar result was observed in New Zealand and Germany, somewhat better in the case of Canada and The Netherlands and the highest, although still of medium strength, in Japan [36,37,38,39,44].

Assigning priority to efficient programmes for food subsidisation and the distribution of food aid has a high potential to contribute to reducing the existing inequalities in health [45]. In Poland, there are a number of initiatives being implemented to address the issue of inequalities in health, but none of them is focused on a particular cause, neither nutrition nor any other subject. Health inequalities are being addressed in a holistic manner, which raises concerns over the extent to which the actions that have been taken may result in any concrete outcomes [46].

In our study, the experts pointed out that tools for assessing the impact of non-food policies on the healthy food environment are lacking. An example in this respect is one of the flagship social policy projects implemented by the Polish government since 2016, i.e., the introduction of a new category of financial benefits for parents raising children. Both in the regulatory impact assessment and in the debate preceding its introduction, all impacts on the food environment have been narrowed down to reducing the scale of poverty and the need for government nutrition programmes. There was no reference to issues such as the impact on food prices and on the consumption of unhealthy food [47]. The problem can also be seen when non-food policies interfere with food-oriented policies. For example, a tax on sugar-sweetened beverages was introduced with the aim of limiting the consumption of unhealthy products. At the same time, the simplified value-added tax (VAT) matrix, which was introduced in the same year but with the aim of achieving public finance goals rather than improving the healthy food environment, allowed for a reduction in the VAT on biscuits and confectionery products [48]. It can be concluded that while the “health in all policies” rule and sustainable development goals are being declared and formally applied in Poland, the actual impact on the health assessment is only apparent.

The priority actions identified in this study tend to be consistent with the identified implementation gaps, such as high deficiencies in relevant domains. Notably, however, actions that received the highest priority scores corresponded to subdomains that had already been assessed as relatively strong. It may be concluded that the expert panel members were leaning towards a strategy of improving good policies that existed in the first place rather than strengthening weak policies or implementing new policies (Figure 3).

The priority actions with the highest priority were those related to food labelling and to creating a system of food-related information policies. Food labelling was given a high priority in the achievability dimension, which is in line with the experiences of other countries. Such a system does not require substantial financial outlays and provides rapid and measurable effects both among adults and among children and adolescents [49].

The lack of a comprehensive informational strategy related to nutrition and food policies manifested after the implementation of the regulation on food sales at school in 2015, resulting in a discussion on the legitimacy of such arrangements, their feasibility, and economic, health, and educational consequences. The greatest controversies were the limitation of the use of salt and sugar in the food products sold at schools, as well as specific indications regarding the size of packaging, product quality, and the portioning of fruit and vegetables, which required producers, wholesalers and franchisees to implement new solutions and look for new suppliers [50]. The result was strong public resistance, followed by a high politicisation of the issue and, ultimately, a far-reaching loosening of the stringency of the policy a year later. The experts participating in our study found the restoration of the original restrictive regulation among the most prioritised actions, highlighting the need to broaden the issue, especially in terms of regulating the place and time of eating a meal during school breaks.

In the case of infrastructure, our study found that the key priority actions, in general, correspond with the gaps identified in the healthy food environment evaluation (see Figure 4). The proposed actions are primarily the implementation of healthy nutrition training for people responsible for children’s nutrition, creating a system for promoting healthy eating habits in the media and covering dietitian counselling in public health insurance. These results correspond to the gaps identified by the Food-EPI tool and are also aligned with the position of the European Economic and Social Committee (EESC) [51], which notes rising concerns about the lack of consumer information on the environmental and social impact of food. These objectives are also included in “Delivering on EU Food Safety and Nutrition in 2050” [52].

The participating experts tended to place great emphasis on issues related to health education. Since the area of education was not included in the original Food-EPI, the proposed actions were linked to the leadership domain as having potentially the highest impact in this area. Educational issues were addressed by the study participants in several ways, showing the general high priority of this issue. Their suggestions are in line with priority projects identified in local communities in Poland in recent years. The small number of recipients of those programmes [53] confirms the identified infrastructure-related gap in the healthy food environment in Poland.

The experts stated that nutritional education should also be present in popular media, including the use of influencers and celebrities to promote healthy choices. This finding remains consistent with suggestions for developing new forms of social communication and translating scientific knowledge, including the celebritisation of science as a solution to reverse the trend of the celebritisation of pseudoscience. There is evidence that this kind of solution might increase the effectiveness of healthy food-promoting activities and attract a wider audience [54].

Incorporating dietitian counselling into the package of basic services financed by public health insurance was the action with the third highest priority in the infrastructure-related domains. Currently, in Poland, dietitian services are not covered by public health insurance, which is similar to most other developed countries. The rationale behind the postulated action is mainly the significantly higher effectiveness of dietitian counselling in terms of changing eating behaviours, as evidenced in a number of studies [55].

The main strength of our study is the fact that it is the first comprehensive and systematic evaluation of Polish food environment policies. We have developed specific recommendations with regard to strengthening the food environment and eliminating the most important gaps in the national food policy system, which have been subject to multidimensional assessment. However, our study is not free of limitations, which result from both the study procedure and the characteristics of the tool we used. One of the basic limitations is the relatively small number of experts who decided to take part in the study. The results we obtained would more precisely reflect the views prevailing in the expert community if the participation rate was higher, especially in the case of the step for the identification and prioritisation of actions. The relatively low level of uniformity of the experts’ responses is also to be noted. A higher participation rate might also improve the credibility of the results obtained, although it seems that any shifts in the assessment results, in light of what has been included in the evidence document, would modify the results on a small scale only. The problem of a small number of participants in the study was limited to some extent due to the specific nature of the study group, which were experts, and also due to the fact that part of the study was qualitative.

Another limitation is the arbitrary method of selecting the international best practices to constitute a benchmark for the assessment of the national food environment. Because assessing the effectiveness of the implemented solutions is possible only in the long term, such an approach appears inevitable; however, for the purpose of future research using the Food-EPI tool, creating a catalogue of practices with well-evidenced effectiveness seems justified. Moreover, the lack of international best practices identified in the case of the “monitoring NCD risk factors and prevalence” indicator prompted its exclusion from the study, which to some extent reduces the level of comprehensiveness of the assessment of the Polish food environment.

Some limitations result from the degree of complexity of the questionnaire that the experts completed. This factor could have been among the main reasons for the low participation rate of the invited experts, and it also could have contributed to the discrepancies in the responses to individual questions. It seems advisable to conduct the study in the form of a workshop, that is, in direct contact with the participants, to reduce some unfavourable phenomena occurring in the course of the study. Although we did indeed conduct a workshop, the limitations resulting from the outbreak of the COVID-19 pandemic became limitations independent of the researchers and the tool used.

Since the food environments are potentially subject to dynamic changes resulting from political decision-making processes, the reported findings might become partially obsolete between the time the study was conducted and the date of the publication. Nonetheless, to our best knowledge, no relevant action was taken to improve the healthy food environment in Poland in the months following the survey. The exception is the sugar-sweetened beverages tax, which started operating in January 2021. The COVID-19 pandemic became a national health policy priority in the period after this study, resulting in the slowing down of policy developments in other areas. 

Finally, the lack of agreement between the experts with regard to the inequality aspect is a limitation, which most likely shows the difficulty in decisively assessing the impact. Different ways of understanding the socio-economic aspect, where it might be understood as the impact on disposable income and welfare, but also as referring to the issue of health inequalities between socio-economic groups, might be one of the reasons for this result. This issue should be subject to improvement in future Food-EPI studies.

## 5. Conclusions


The healthy food environment in Poland has been assessed as being very weak or weak in most cases of the indicators related to policies, with only three indicators being assessed as strong policies. The largest gaps were identified in the food retail domain, along with individual indicators related to other domains, namely, restricting unhealthy food promotion to children on social media, restricting unhealthy food promotion to children on packaging, healthy public procurement standards and support and training systems in private companies. Infrastructure was assessed as being slightly better, with more indicators being assessed as moderate policies and four indicators being assessed as strong policies. The main gaps identified here are the political support related to priorities for reducing health inequalities, the use of evidence in food policies, transparency in the development of food policies and assessing the public health impact of non-food policies.Taking into account the combined dimensions of importance, achievability and equity for policies and importance and achievability for infrastructure, the most prioritised actions identified to be implemented to improve the Polish health food environment are the implementation of a clear and simple labelling system for food products, including information on salt/sugar/trans fats, and the implementation of a system of training on healthy eating rules targeting people responsible for feeding children. These constitute the basic recommendations regarding food policy actions to be implemented by the public authorities in Poland.


## Figures and Tables

**Figure 1 foods-11-01648-f001:**
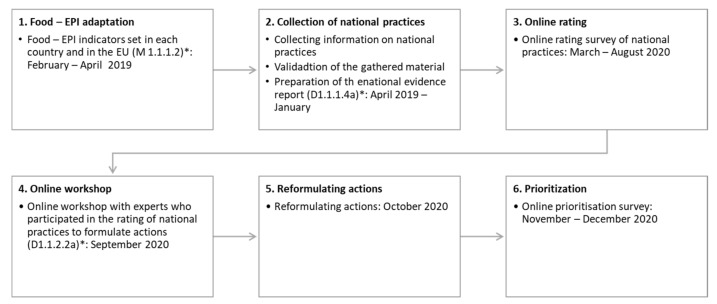
Steps of the Healthy Food Environment Policy Index applied in this study (2019–2020) to assess the strength of EU policies and identify priority actions. * Milestones (M) and deliverables (D) are in accordance to the PEN project.

**Figure 2 foods-11-01648-f002:**
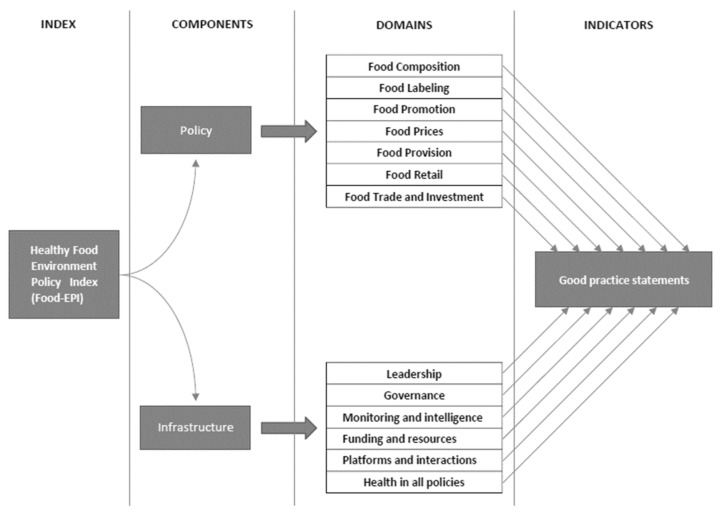
Healthy Food Environment Policy Index components, domains and indicators.

**Figure 3 foods-11-01648-f003:**
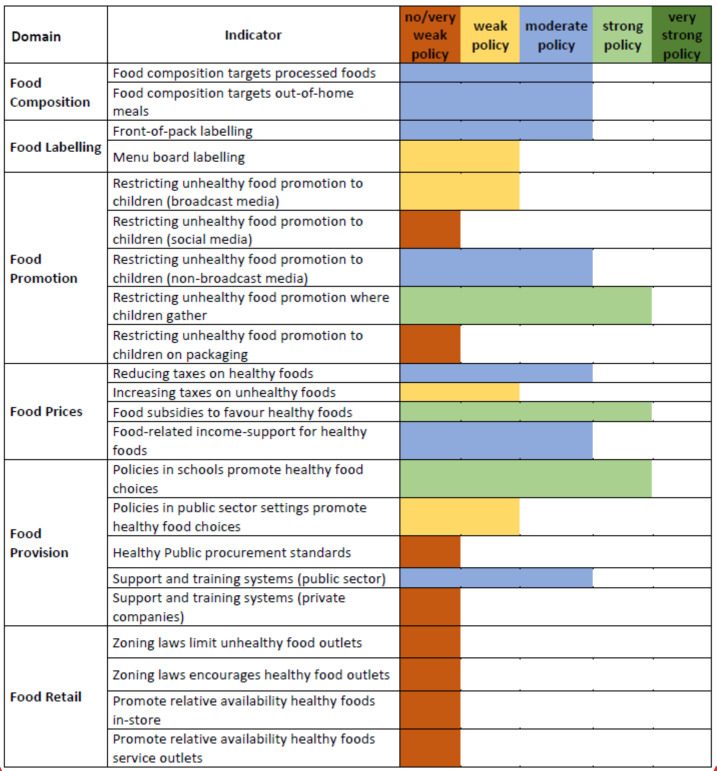
Results of the implementation of food-related policies in Poland across the Food-EPI domains and indicators.

**Figure 4 foods-11-01648-f004:**
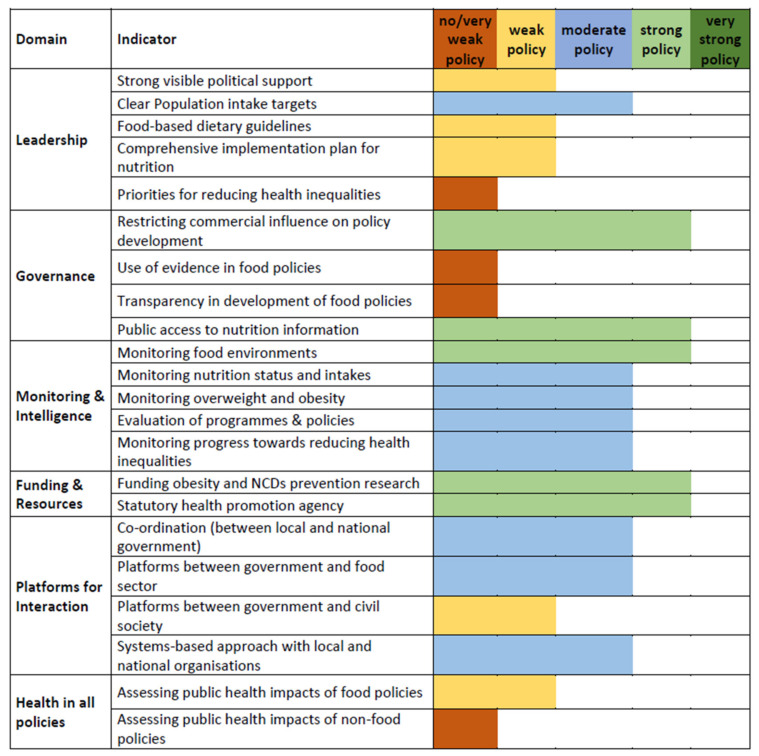
Results of the Food-EPI evaluation in Poland for the infrastructure-related domains and indicators.

**Figure 5 foods-11-01648-f005:**
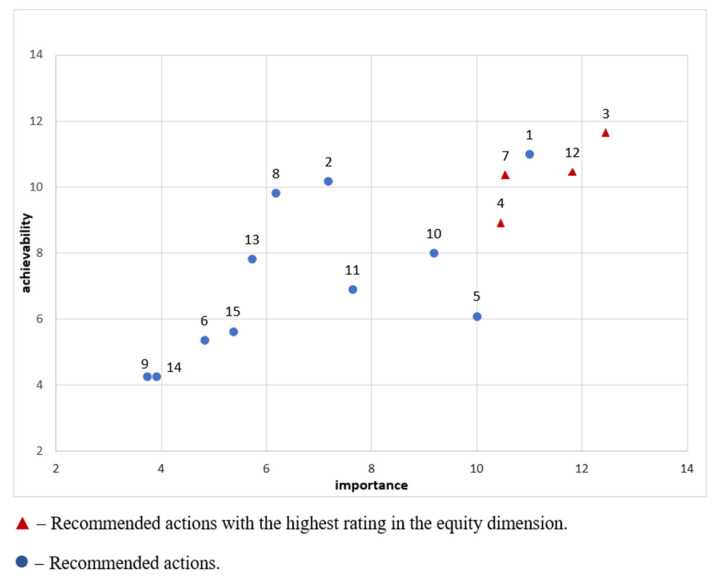
Recommended policy actions with details regarding importance and achievability (no weighting). The numbers in the diagram have been determined in accordance with the information on the numbers appointed to recommended actions (see Table 1).

**Figure 6 foods-11-01648-f006:**
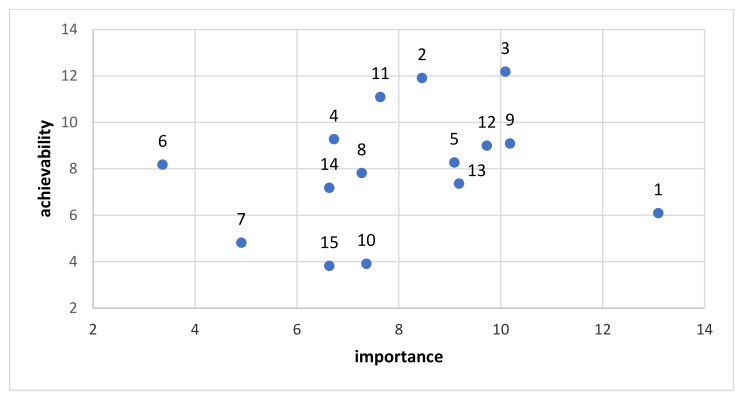
Recommended infrastructure support actions with details regarding importance and achievability (no weighting). The numbers in the diagram have been determined in accordance with the information on the numbers appointed to recommended actions (see Table 2).

**Table 1 foods-11-01648-t001:** Policy-related actions recommended for improving the Polish healthy food environment, ranked by priority. The action labels (action 1, action 2, …) are taken from the original document sent for prioritisation to experts.

Rank	Action	Domain/Indicator in the Food-EPI That the Respective Action Refers to	Average Score (Importance, Achievability, Equity Scores Combined, Weighting Included)
**1**	Action 3. Introduce a clear and simple labelling system for food products, including information on salt/sugar/trans fats.	2/1	4.0
**2**	Action 12. Prepare the launching of information campaigns that are thoroughly prepared from the sociological and psychological point of view, preceding the introduction of food policy regulations.	All	3.69
**3**	Action 1. Modify the Ordinance of the Minister of Health on groups of foodstuffs intended for sale to children and adolescents in education system units and the requirements to be met by foodstuffs used as part of mass nutrition for children and adolescents in these units in a way that specifies requirements that are consistent with nutritional requirements and recommendations.	5/1	3.57
**4**	Action 7. Modify school curricula by adding a subject or at least a compulsory thematic block on “nutritional education”.	5/2 5/4	3.53
**5**	Action 5. Change the VAT matrix in a way that unequivocally promotes low-processed foods and healthy food choices.	4/1	3.21
**6**	Action 4. Introduce warnings on food products and/or during advertisements for products that are not recommended in the daily diet (e.g., sweets, energy/sweetened drinks) in line with regulations regarding drugs or tobacco product packaging. Regulate the media market in the field of unhealthy food advertisements, including the application of consistent regulations to the entire media market, including the internet.	2/1 3/1 3/2	3.14
**7**	Action 2. Regulate the rules for eating meals in schools with regard to the time and place of eating a meal (reserve enough time for and the regularity of eating meals).	5/1	3.03
**8**	Action 10. Introduce regulations limiting the exposure of unhealthy food in commercial establishments, especially in regard to exposing such food to the eyes of children and/or prohibiting displaying such food in the vicinity of cash desks.	6/1	2.98
**9**	Action 8. Introduce a requirement to label menu items in restaurants, taking into account the nutritional and energy value of the dishes served.	2/2	2.70
**10**	Action 13. Introduce food fortification (e.g., with folic acid) to eliminate the most common shortages.	1/1	2.44
**11**	Action 6. Modify the assumptions of the nutritional policy implemented in relation to poor individuals in a way that promotes healthy eating choices (e.g., creating a catalogue of products that cannot be purchased with food vouchers and/or rewarding the purchase of low-processed food).	4/3 4/4	2.39
**12**	Action 11. Implement social impact programmes aimed at building a culture of trust and support for regulations in the field of healthy eating.	All	2.31
**13**	Action 15. Introduce a system of incentives and discounts for small entrepreneurs for running small vegetable and fruit stores to increase access to low-processed food.	6/2	1.85
**14**	Action 14. Introduce legal requirements regarding public procurements in a way that enforces the use of the quality criterion of purchased food products.	5/3	1.69
**15**	Action 9. Introduce a system of controlling the density of fast-food bars in the public space, e.g., through a licensing system, taking into account the distance from educational establishments.	6/1	1.45

Note: The final score reported in the right-hand column is a result of averaging scores attributed by all experts. The final scores were additionally modified by the weights attributed by experts to the dimensions, as described in the methods section.

**Table 2 foods-11-01648-t002:** Infrastructure-related actions recommended for improving the Polish healthy food environment, ranked by priority. The action labels (action 1, action 2, …) are taken from the original document sent for prioritisation by experts.

Rank	Action	Domain/Indicator in the Food-EPI That the Respective Action Refers to	Average Score (Importance, Achievability, Equity Scores Combined, Weighting Included)
**1**	Action 3. Introduce a system of trainings on healthy eating rules targeting people responsible for feeding children (including cooks, authorizing officers, parents).	2/4	5.67
**2**	Action 2. Promote the principles of healthy eating using marketing tools, media campaigns and influencers.	2/2	5.21
**3**	Action 1. Introduce reimbursed dietitian services at the level of primary health care and specialist care.	1/5 2/4	5.1
**4**	Action 9. Legally regulate the profession of dietitian.	2/3	4.83
**5**	Action 11. Facilitate the availability of fruit and vegetables in schools and workplaces.	¼	4.7
**6**	Action 12. Introduce dietary supervision of nutrition in educational institutions and child care and caregiving institutions (e.g., require a special administrative position at the municipal/commune level).	1/2 3/2	4.67
**7**	Action 5. Creating leaders—the implementation of educational programmes to disseminate nutritional knowledge among people responsible for the education and upbringing of children and the introduction of an appropriate educational module as part of pedagogical studies.	1/1	4.35
**8**	Action 13. Introduce systemic and transparent rules of collective nutrition in public entities (especially hospitals) in accordance with evidence-based public health. Introduce hospital nutrition education.	2/2	4.25
**9**	Action 4. Create leaders who promote healthy food choices and support public action at various levels, including the micro-level (e.g., school managers).	1/1	4.19
**10**	Action 8. Introduce tax solutions that support employers in promoting healthy food choices in the workplace.	1/5	3.86
**11**	Action 14. Introduce unified requirements regarding the need to conduct evaluations of implemented public health activities.	3/4	3.58
**12**	Action 6. Introduce a public communication system for sanitary inspection alerts regarding food products and dietary supplements (e.g., a system analogous to the alerts of the Government Centre for Security distributed via mobile telephone networks).	3/1	3.2
**13**	Action 10. Implement regularly conducted epidemiological studies and monitoring of the nutritional status of the population as a basis for the implementation of public health actions.	4/1 2/2	2.86
**14**	Action 15. Create a system of information exchange and coordination of activities of entities implementing nutrition policy tasks, health education and health promotion to increase the effectiveness of spending funds and to improve measures targeting the population’s health status, especially its nutritional status.	5/4	2.86
**15**	Action 7. Introduce clear and transparent solutions for food certification to build public confidence in this system. Develop the ability to recognize quality certificates.	2/1 2/3	2.43

Note: The final score reported in the right-hand column is a result of averaging scores attributed by all experts. The final scores were additionally modified by the weights attributed by experts to the dimensions, as described in the methods section.

## Data Availability

Full report from the research available at: https://www.jpi-pen.eu/reports.html (accessed on 3 May 2022).

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
