# Peer review of "The Healthy Food Environment Policy Index in Poland: Implementation Gaps and Actions for Improvement"

_foods, 2022, doi:10.3390/foods11111648_

Round 1
Reviewer 1 Report
Thank you for the opportunity to review this manuscript. This manuscript could benefit from some extensive editing to improve the clarity and rational for the presented work. Specific suggestions are outlined below.
Abstract
Lines 24 – 25 Please specify what the ‘Polish policies and infrastructure’ were related to, and what the ‘list of recommended actions’ were intended to do.
32 – What are these indicators of? Please specify what the scoring system indicates.
Introduction
51 - What are some of these ‘solutions in regard to food-based dietary guidelines’? Please explain.
53 - What is the response of Polish policy to and how is it insufficient? Please elaborate.
Methods
Please describe the rationale for why the Food-EPI tool was used in this study.
Please provide column headings in English for the Supplementary Materials Evidence Document. It may be helpful to include an additional sheet explaining the data labels and where the data was obtained from in the document.
In Section 2.1.3 (starting on line 116), please specify what language this data was collected in and if and how it was translated to English.
Include an ethics statement for research involving human subjects as outlined in the author guidelines: https://www.mdpi.com/journal/foods/instructions
Results
Figures 3 & 4 – What do the length of the colored bars represent? The figures should be revised to clarify the meaning of the bar length.
Please specify that the column headings are Food-EPI domains and indicators.
244 – 245 Why was this result ‘tended to be expected’? This statement is unclear and should be revised.
Tables 1 & 2- Please include a footnote or other indicator to outline the maximum score possible for values listed in the right column.
Why are the actions in Tables 1 & 2 are presented by average score? It is somewhat confusing to see the action numbers out of order in the left column. Please clarify in the text and Table Title that actions are listed from highest to lowest score or revise the tables to list the actions in order of their number.
335 – 337 It appears the authors have left text from the manuscript template instructions. Please remove this.
Discussion
362 – 363 Please clarify what the ‘expectations on the consumer side’ are for.
401 – What is meant by ‘only the actual impact on health assessment is apparent.’? Please clarify.
481 – 482 What are the different “ways of understanding the socio-economic aspect”? Socio-economic aspects of what? Please revise this statement.
Reviewer 2 Report
Dear Authors,
Thank for this interesting paper.
However, there are some points to improve:
(i) The Abstract does not have conclusions, the project should also be mentioned there;
(ii) The objectives of the study in the Introduction do not match with those in the Abstract. Please, check it;
(iii) The methods are not clearly explained. Please, provide more detailed explanation of the survey (questions, criteria for the respondents, where it took place, etc.);
(iv) The results are mixed with discussion. Please, stricktly describe the results, give the data of the respondents, move the explanation of the results to the discussion;
(v) The Conclusions should address the objectives.
Please, rework it.
(vi) The text needs language proof-reading.
I also think that the data presented in the paper (2020) are a bit too old and do not reflect current food policy in Poland. It would be nice to include some reflections on this in the text.
Good luck with your paper!
Round 2
Reviewer 1 Report
The authors have addressed my comments and I would like to wish them the best of luck in their future endeavors.
Author Response
Dear Reviewer,
We would like express our gratitude for your valuable remarks in the first round. Since no further comments were provided in the second round, no responde is needed from our side.
Reviewer 2 Report
Dear Authors,
Thank you for the updated version.
(i) The abstract has been revised. However, there is still need to improve the language. "Food environment" appears too often in the text.
(ii) There are the issue with punctuation throughout the text: i.e., . [1,2]; ,(14)... etc. The British style was also not used: prioritiZation, synthetiZed... etc.
(iii) The aims in the abstract and Introduction are different. It sounds confusing. Please, modify.
(iv) The discussion is too focused on the Polish case. Please, extend it with international best practices (they are somehow mentioned in the supplementary) and include references to international publications in high-ranking journals.
(v) Lines 539-549 should be moved to the beginning of Discussion and well referred.
(vi) Lines 550-557 sound as a critical methodological gap.
(vii) Please, replace the numbers with words throughout the text: i.e., 3 - three (line 561), etc.
